# R&D Performance Evaluation in the Chinese Food Manufacturing Industry Based on Dynamic DEA in the COVID-19 Era

Shiping Mao [1], Marios Dominikos Kremantzis [1,*], Leonidas Sotirios Kyrgiakos [2] and George Vlontzos [2]

1   University of Bristol Business School, University of Bristol, Howard House, Queens Avenue, Bristol BS8 1SD, UK
2   Department of Agriculture Crop Production and Rural Environment, School of Agricultural Sciences, University of Thessaly, Fytoko, 38446 Volos, Greece
*   Correspondence: marios.kremantzis@bristol.ac.uk; Tel.: +44-1174558351

**Abstract:** Nowadays, China's food consumption structure is shifting from being survival-oriented to health-oriented. However, the food industry is still facing a research and development (R&D) dilemma. Scientific evaluation of an enterprise's R&D performance can help to reduce the investment risk of R&D and promote economic benefits. This study implements the dynamic data envelopment analysis (DDEA) technique to measure and evaluate the level of R&D performance in the Chinese food manufacturing industry. Twenty-eight listed companies were selected for the study, considering the time period from 2019 to 2021. After constructing a system of inputs, outputs and carry-over indicators, overall and period efficiency scores were obtained. The results reveal that the overall level of R&D in the industry is relatively low (0.332). Average efficiency scores across years were estimated as 0.447, 0.460, 0.430 for 2019, 2020, and 2021, respectively. Lastly, this study considers the actual business situation of the industry and makes suggestions for improvement from the perspective of enterprises and the government; these anticipate aiding the food manufacturing industry to improve the performance management of R&D activities.

**Keywords:** dynamic DEA; research and development; food; efficiency

## 1. Introduction

It is widely accepted that the food industry is related to people's livelihoods and is essential for economic development. According to the Food and Agriculture Organisation of the United Nations (FAO), if the population reaches 9.1 billion by 2050, global food production will need to increase by 70% [1]. China, one of the four largest economies in Asia, has seen a steady growth in the market size of its food industry. As the epidemic recovers, China's food manufacturing revenue is expected to reach approximately $339 billion by 2025 [2]. In addition, the emergence of a wealthy society has resulted in shifting consumers' needs from simple sustenance to diverse, high quality, and sustainable products, creating opportunities and challenges for the food industry.

Research and development (R&D) is a decisive factor in achieving a competitive advantage for businesses [3]. Nevertheless, in 2020, the food manufacturing sector was connected with 11,560 R&D projects, which is only around one-seventh of the manufacturing industries with the highest number of R&D projects [4]. Additionally, the food industry invests much less in R&D than other industries, with only 1/20th of the highest [5]. The way of enhancing R&D capabilities to meet the current needs of the market and thus provide an incentive for Chinese food companies should be further assessed. Therefore, it would be meaningful to establish an objective performance evaluation system, in order to identify and analyse the problems in the food R&D process and, thus, propose suggestions for improvement.

### 1.1. Evaluation of R&D Performance

The increasing complexity of R&D activities and the large number of R&D resources currently available act as a restricting factor for companies to conduct performance evaluations. These rationalise the use of limited resources and thus aid companies to adjust their R&D plans. To face the challenges, scholars have also made attempts using different methods, summarised in Table 1.

**Table 1.** Methods for the evaluation of R&D performance.

| Methods | Sources |
| --- | --- |
| Peer review | Pölönen et al. [6]; You and Jung [7] |
| Bibliometric methods | Ding et al. [8]; Zou [9] |
| Analytic hierarchy process (AHP) | Wang et al. [10]; Shin et al. [11] |
| Balanced scorecard (BSC) | Bigliardi and Ivo Dormio [12] |
| Factor analysis | Guo and Yang [13] |
| Data envelopment analysis (DEA) | Chachuli et al. [14]; Chen et al. [15] |
| Stochastic frontier analysis (SFA) | Wang [16]; Matricano et al. [17] |

To the best of our knowledge, the current literature mainly focuses on static R&D efficiency over a specific period. The approach of using a dynamic perspective to measure the interrelationship role in the R&D process has not yet been widely explored. In fact, the company's investment in R&D is cumulative, affecting an extended time period [18]. To be more precise, the output of one period operates as an input for the next one. A dynamic framework allows the real-time allocation of resources to follow changes over time, avoiding errors in efficiency measurement [19]. Kao [20], for instance, used the case of the Taiwanese forests to demonstrate that a static assessment of DMUs only leads to overestimating efficiency. In this case, the forest stock is used as a quasi-fixed input linking two consecutive periods, and the dynamic analysis results are more accurate and reliable than the ones obtained via the static analysis. Therefore, an evaluation of the overall system and period efficiency of R&D inputs over multiple periods could be essential.

In addition, current research focuses, at the macro level, in the area of R&D and national economic growth [21,22] and comparisons of R&D efficiency between countries [23,24]. However, there are few assessments of the R&D performance of industries or companies. Data collection due to non-disclosure and the inconsistency in the statistical calibre of each company's R&D data are the main challenges which impede research on this topic [25].

In particular, of the small number of existing industry studies, scholars have focused more on high-tech industries and less on traditional manufacturing [13,18]. Some studies have shown that the food industry has a relatively low R&D intensity compared to other manufacturing industries [25,26]. Gopinath and Vasavada [27] suggest that there are spillover effects from R&D activities in the food industry through empirical analysis. It implies that the R&D achievements of the leaders in the food industry can be easily imitated by other firms, which further undermines firms' enthusiasm to invest in R&D. Nevertheless, the evaluation of the industry's R&D performance is still critical. The food industry is on a constant quest to fulfil consumer needs, such as additive-free, organic, and sustainable food, and therefore R&D and innovation are necessary.

Overall, the methods used for the evaluation of the R&D performance activities are very different. They have been applied mainly to macro or high-tech industries, with less emphasis on traditional manufacturing industries such as the food industry. However, the specifics of its R&D efficiency are not sufficiently clear for the food processing industry. Moreover, the vast majority of research on performance assessment is based on a static perspective, and the industry's inter-relationship between the multiple years has not been

explored. Considering intertemporal effects helps companies to allocate R&D resources appropriately over time.

*1.2. Dynamic DEA*

The use of Window analysis [28] and the Malmquist index [29] was the first attempt by scholars to investigate multi-year efficiency. However, this approach fragmented the individual periods and ignored the presence of carry-over activity between periods [30]. The dynamic DEA approach is deemed as a good remedy for this shortcoming. Dynamic DEA, which was originally introduced by Färe and Grosskopf [31], is an improved DEA model designed to measure the relative efficiency of DMUs with a dynamic perspective over time. Following this, scholars have made significant methodological improvements towards a more compact application of a dynamic DEA approach to help solve real-world problems. Table 2 summarises the development and applications based on key literature.

Mariz et al. [32] reviewed the application of dynamic DEA during the past 30 years. They found that the energy, environment, and transport sectors were the most prevalent ones, accounting for 15%, respectively. In addition, studies using dynamic DEA are concentrated in the USA, China, and Taiwan. Little attention, however, has been paid to the performance evaluation in the traditional food manufacturing industry using a dynamic DEA model, regarding R&D efficiency improvements.

**Table 2.** Methods for the evaluation of R&D performance under the dynamic DEA techniques.

| Author & Year | Area Identified | Techniques |
| --- | --- | --- |
| Chen [33] | Production network | Dynamic Network—DEA model (DNDEA) |
| Tone and Tsutsui [30] | Electric utilities in US and Japan | Slacks-based dynamic DEA model (DSBM) |
| Tone and Tsutsui [34] | Electric power companies | Dynamic Slacks-based model with network structure (DNSBM) |
| Jafarian-Moghaddam and Ghoseiri [35] | World's railways | Fuzzy dynamic multi-objective DEA model |
| Kao [20] | Taiwanese forests | Dynamic relational analysis model |
| Omrani and Soltanzadeh [36] | Airlines in Iran | Dynamic network DEA model |

In this study, we measure and evaluate the R&D performance of the Chinese food manufacturing industry across multiple years and multiple sectors, by applying a relational dynamic DEA approach, proposed by Kao [20]. This study provides a macro and comprehensive picture of the current situation of the R&D performance of the Chinese food industry, while proposing further improvements for resource allocation and avoiding unnecessary waste of human, material, or financial resources; this will, thus, improve the efficiency of R&D abilities. Finally, it can facilitate policymakers to implement a scientific program of technological advancement that will increase economic contribution through enhanced levels of R&D performance.

The remainder of the paper is organized as follows. Section 2 briefly describes the research methodology followed, by putting more emphasis on the DEA approach and the dynamic DEA model based on Kao's study [20]. This section also presents the framework for constructing input and output indicators for R&D in the food industry and the potential logic for selecting indicators. Next, Sections 3 and 4 showcase the results and relevant discussion, respectively. Finally, Section 5 presents conclusions and future research directions.

**2. Methodology**

Data envelopment analysis (DEA) is the primary analytical method used in this study. Compared to other methods of performance assessment, DEA has several advantages: for example, the DEA approach enables the evaluation of a system that makes use of (multiple) inputs to produce (multiple) outputs. In addition, there is no need to make assumptions or prioritize weights, making them highly objective [37]. A multi-stage evaluation study on the level of R&D performance of food manufacturing companies using the DEA method

can ensure the objectivity and accuracy of the measurement and evaluation results. In particular, this study draws on Kao's dynamic relational DEA model [20] to explore the topic. This section demonstrates the study's design by explaining the construction of the dynamic DEA model, the selection of operational factors, and the data collection and analysis procedures.

### 2.1. Model Specification

The dynamic relational model, proposed by Kao [20], enables the relative efficiency of the overall system and its individual periods to be measured separately over multiple periods. Unlike the inputs, the quasi-fixed inputs take a while to adjust to the optimum level [38]. In this case, quasi-fixed input is forest stock as it operates mainly in the sense that the forest trees left after harvesting in one period can continue to grow in the next. Quasi-fixed input is used as the flow to link two adjacent periods in series, allowing the several periods to possess interlinkages. The model optimises and improves on the relational model proposed by Kao and Hwang [39]; the model proposed in 2010 does not consider the presence of intermediate products in the first and last periods, thus compensating for the shortcomings. In addition, overall efficiency and period efficiency can be calculated simultaneously, and the mathematical relationship between the two is explored as well, thus contributing more to the understanding of the relationship between the whole and its constituent parts. The dynamic system used in this study refers to Kao's relational model [20] to calculate the R&D efficiency scores of the Chinese food manufacturing industry over the last three years.

Cook et al. [40] argue that knowledge of the performance measurement objectives can influence the model's orientation, which means understanding whether the industry achieves an efficient production frontier by reducing inputs or expanding output. Food research and development aim to develop new products and place them on the market to meet consumers' needs [41]. The management of food manufacturing companies expect more R&D investment to attain higher levels of profit; this implies an output enhancement and thus, an output-oriented model has been chosen for this study.

In order to measure the relative efficiency of n DMUs, let $X_{ij}^{(t)}(i = 1, 2 \ldots m)$, $Y_{rj}^{(t)}(r = 1, 2 \ldots s)$ and $Z_{fj}^{(t)}(f = 1, 2 \ldots g)$ donate the ith input, rth output and fth flow, respectively, to the jth DMU in period $t$. $Z_{fj}^{(t)}$ plays a linkage role between two consecutive periods, which means $Z_{fj}^{(t)}$ acts not only as an output in period $t$, but also as an input in the next period $t + 1$. In addition, donate $X_{ij} = \sum_{t=1}^{p} X_{ij}^{(t)}$ and $Y_{rj} = \sum_{t=1}^{p} Y_{rj}^{(t)}$ as the total quantities of the $i$th input and $r$th output within all $t$ periods, respectively [20]. Figure 1 demonstrates the general dynamic system used in this study, which connects two adjacent periods with $Z_{fj}^{(t)}$.

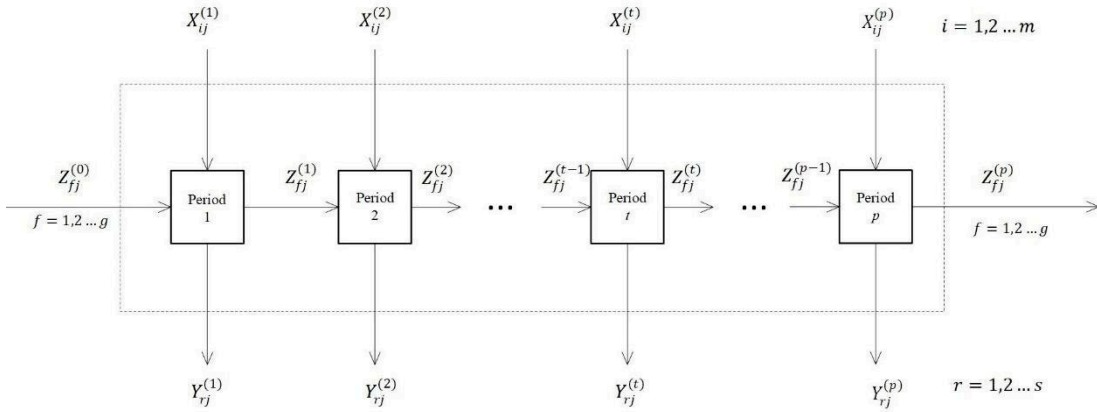

**Figure 1.** A general framework of the dynamic system with carry-overs. Adapted from [20].

According to the structure in Figure 1, the output-oriented model is developed to measure the efficiency of $DMU_k$ under the assumption of constant returns to scale with Kao's relational model [20]:

$$\max \frac{1}{E_k} = \sum_{i=1}^{m} v_i X_{ik} + \sum_{f=1}^{g} \omega_f Z_{fk}^{(0)}$$

Subject to:

$$\sum_{r=1}^{s} u_r Y_{rk} + \sum_{f=1}^{g} \omega_f Z_{fk}^{(p)} = 1$$

$$\left( \sum_{i=1}^{m} v_i X_{ij} + \sum_{f=1}^{g} \omega_f Z_{fj}^{(0)} \right) - \left( \sum_{r=1}^{s} u_r Y_{rj} + \sum_{f=1}^{g} \omega_f Z_{fj}^{(p)} \right) \geq 0 \tag{1}$$

$$\left( \sum_{i=1}^{m} v_i X_{ij}^{(t)} + \sum_{f=1}^{g} \omega_f Z_{fj}^{(t-1)} \right) - \left( \sum_{r=1}^{s} u_r Y_{rj}^{(t)} + \sum_{f=1}^{g} \omega_f Z_{fj}^{(t)} \right) \geq 0, \ j = 1, 2 \ldots n; t = 1, 2 \ldots p$$

$$u_r, \ v_i, \ \omega_f \geq \varepsilon, \ r = 1, 2 \ldots s; \ i = 1, 2 \ldots m; \ f = 1, 2 \ldots g$$

where $v_i$, $u_r$, $\omega_f$ are virtual multipliers of input factor, output factor and carry-over, respectively. The $\varepsilon$ is a small non-Archimedean number, which was introduced to avoid ignoring the metric because of the "zero weight" [42].

By solving this linear programming model, the optimal solution $\left( u_r^*, v_i^*, \omega_f^* \right)$ is obtained from model (1). Furthermore, the efficiency of the whole system, $E_k^s$ and the efficiency of the period t, $E_k^{(t)}$ can be calculated using Formulae (2) and (3), respectively:

$$E_k^s = \frac{\sum_{r=1}^{s} u_r^* Y_{rk} + \sum_{f=1}^{g} \omega_f^* Z_{fk}^{(p)}}{\sum_{i=1}^{m} v_i^* X_{ik} + \sum_{f=1}^{g} \omega_f^* Z_{fk}^{(0)}} \tag{2}$$

$$E_k^{(t)} = \frac{\sum_{r=1}^{s} u_r^* Y_{rk}^{(t)} + \sum_{f=1}^{g} \omega_f^* Z_{fk}^{(t)}}{\sum_{i=1}^{m} v_i^* X_{ik}^{(t)} + \sum_{f=1}^{g} \omega_f^* Z_{fk}^{(t-1)}} \tag{3}$$

### 2.2. Selection of Operational Factors

The selection of inputs, outputs, and carry-overs (flows) indicators for the dynamic model in this study is based on the existing literature, so as to collect the appropriate data for the Chinese food supply sector. Table 3 summarises the existing literature based on the choice of input and output indicators for R&D performance in various industries.

Regarding input indicators, most scholars choose indicators from the perspective of human, material, and financial resources [19,43,44]. Of these, R&D expenditure and R&D labour are two commonly used indicators to measure R&D investment [44]. R&D expenditure includes the salaries of R&D personnel, raw materials consumed, and depreciation of assets used in the research process. R&D labour represents the number of full-time R&D personnel in a company. These two indicators reflect the importance companies attach to R&D and their ability to innovate independently [45]. Thus, they are selected as inputs for this study.

For food manufacturing companies, it is imperative to turn R&D-type investments into economic benefits so that more profits can be made. Table 3 also indicates that revenue and profits are the mostly used output indicators. Revenue is divided into operating revenue and non-operating revenue. Since the non-operating revenue is not directly related to daily operating activities, its inclusion may lead to imprecise results in measuring R&D performance. Therefore, operating revenue is chosen as an output indicator in this study. In addition, net profit is the ultimate result of a business's operations and has therefore been chosen as an output indicator.

**Table 3.** Indicator selection for R&D performance evaluation.

| Related Literature | Industry | Input | Output |
|---|---|---|---|
| Liu et al. [44] | Industrial enterprises | R&D expenditure<br>R&D personnel | Sales revenue of new products<br>Revenue from principal business |
| Delvaux et al. [25] | Food-processing firms | Total cost of goods sold<br>Physical capital<br>Labour | Revenue |
| Chachuli et al. [14] | Renewable energy industry | Publication<br>Human capital<br>Patent | Installed capacity |
| Xiong et al. [19] | Research institutes | R&D expenditure<br>R&D personnel | Income |
| Yu et al. [18] | High-tech industry | R&D expenditure<br>R&D labour | New product revenue<br>Revenue |
| Wang et al. [43] | New energy enterprises | Fixed assets<br>Staff wages<br>R&D costs | Total profits<br>Market value |
| Hashimoto and Haneda [46] | Pharmaceutical industry | R&D expenditure | Patents<br>Pharmaceutical sales<br>Operating profit |
| Schmidt-Ehmcke and Zloczysti [47] | Manufacturing industry | R&D investments | Patent applications |

The capital stock is a dynamic factor with a continuum [48]. The capital stock can represent the capital resources available to a food manufacturing company and reflects the scale of the company's production operations in a given year. This implies that the capital stock can be considered as the current year's output. However, in another sense, it is the sum of the various types of capital invested in a food manufacturing company in the following year. It can therefore be considered as an input in the following year. Therefore, capital stock is engaged in inputs and outputs and can be a carry-over (also called intermediate product) factor to link up two adjacent periods. Table 4 summarises the indicators selected for the study. All indicators are measured in CNY, except R&D labour, which is measured in terms of headcount.

**Table 4.** Measure indicators of the R&D process.

| Role | Indicator | Description | Data Source |
|---|---|---|---|
| Input | R&D expenditure | Direct expenses related to the company's efforts to develop, design, and enhance its products or technologies | Annual reports and financial statements |
| Input | R&D labour | The number of full-time R&D personnel in the given year | |
| Carry-over | Capital stock | The total share capital issued by a company | |
| Output | Operating revenue | Revenue generated from the company's primary business activities | |
| Output | Net profit | Total profit of the enterprise for the period after deduction of income tax | |

*2.3. Data Collection*

This study follows the industry classification guidelines issued by the China Securities Regulatory Commission in 2012 and selects listed companies in the food manufacturing industry (Code C14) under the broad manufacturing category (Category C) as research

subjects. The Chinese R&D food manufacturing industry was selected due to its high amount of spending, which exceeds even that of the USA [49]. All the companies listed are traded on stock exchanges in the territory of China as well [50]. For objective reasons, such as special treatment and the unavailability of R&D data, we failed to utilize the data of all companies and needed to screen these companies. The rationale of this screening is described in more detail below. Furthermore, the time period from 2019 to 2021 was selected for this study. However, due to the carry-over factor, the study needed to include data from 2018 to 2021. This particular period was chosen for the study since companies began to announce R&D information after 2018. Figure 2 visualises the dynamic structure of the indicators applied to this study.

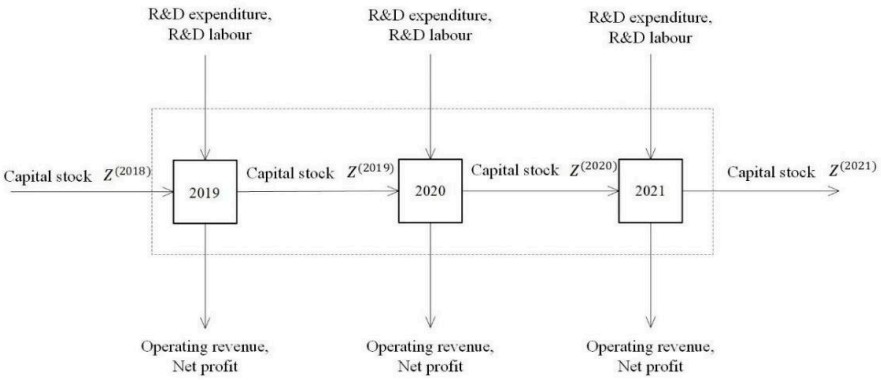

**Figure 2.** Dynamic system of inputs, outputs, and carry-over of DMUs.

The annual financial report reflects essential information on the enterprise's financial position, operating results, and cash flow position for the year. Considering the availability and authenticity of the data of food manufacturing enterprises, this study uses the publicly released financial statements and annual reports with standard audit opinion reports issued by audit firms as the primary data source for the model. The data is collected by visiting the companies' official websites, which is completely legal and does not entail data leakage. To enhance the rigor and validity of the study, the following principles were followed in selecting all the enterprises under consideration.

Firstly, it is more valuable to collect a larger sample of studies for the purpose of evaluating R&D performance; however, a significantly large number of DMUs can reduce the ability of DEA to differentiate [40]. A 'rule of thumb' in DEA-relevant literature is that the number of DMUs should be at least twice the number of input and output combinations [51].

Secondly, listed companies with unusual financial positions or other unusual circumstances in the study period from 2018 to 2021 were excluded. Shanghai and Shenzhen Stock Exchanges have given these companies the special treatment (ST) label. It means that they have been operating at a loss for two or three consecutive years and may have problems with their business management compared to the rest of the sample [52]. Thus, such companies will not be considered to make the study more comparable and informative.

Lastly, indicators with negative values are excluded. One of the assumptions of the DEA approach is that the input and output indicators are non-negative [53]. The data selected may have negative net profit values that do not meet the DEA assumptions and are therefore excluded.

After screening, 28 listed Chinese food manufacturing companies met the above-mentioned requirements and, as such, they were selected to be further analysed with DEA. All inputs, carry-overs and outputs set into the benchmarking process can be found on Appendix A.

*2.4. Results Analysis*

To better understand the data, descriptive and correlation analyses were conducted on operational indicators (inputs, outputs, and carry-over). Then, for modelling, running, and

analysing our data, we utilised the programming language Python 3.7.6 and, in particular, version 2.1 of PuLP as the free linear programming library. The R&D efficiency scores for the whole system and the period efficiencies will be collated. In order to provide a more tangible measure of the overall level of R&D across the food industry, the average R&D efficiency and the corresponding ranking of each DMU were presented. Finally, we used Spearman's rank-order correlation to check for strong correlations between non-parametric measures with ranking variables over a three-year period, which made our analysis more rigorous. Additionally, efficiency differences between sectors and periods were assessed for the examined time period by implementing the non-parametric Kruskal–Wallis test.

## 3. Findings

### 3.1. Descriptive Statistics

Tables A1 and A2 gives an overview of the descriptive analysis of the operating indicators for the observation period from 2019 to 2021. In terms of input indicators, the number of R&D expenditures and R&D labour varies and is highly discrete among food manufacturing companies. In addition, R&D expenditure and the number of R&D labour increased year on year over the last three years, by 34.3% and 33.5%, respectively. Figure 3 presents that food manufacturing companies are aware of the importance of R&D and are beginning to invest more. At the same time, investment in R&D activities has stimulated output. Based on the descriptive data, we find that operating revenue and net profit also show a year-on-year growth trend between 2019 and 2021 (Table A2).

**Figure 3.** Average value of selected indicators per year.

### 3.2. Correlation Analysis

Based only on the descriptive analysis (see Table A2), drawing any specific linkage between R&D inputs and benefit outputs could be frivolous. To this end, Table 5 uses Pearson's correlation coefficients to show the relationships between inputs and outputs in model (1). It displays that input and output indicators are positively correlated at the significant level of 0.01; this indicates that the higher the R&D input, the greater the generated benefits. In particular, we find a strong relationship between the input variable "R&D expenditure" and the two output variables, with correlation coefficients of 0.638 and 0.839, respectively. The following sub-section will show the R&D efficiency scores based on multiple input and output indicators using the dynamic DEA approach.

**Table 5.** Coefficient of correlation between inputs and outputs.

|  | R&D Expenditure | R&D Labour | Operating Revenue | Net Profit |
|---|---|---|---|---|
| R&D expenditure | 1 |  |  |  |
| R&D labour | 0.610 ** | 1 |  |  |
| Operating revenue | 0.638 ** | 0.313 ** | 1 |  |
| Net profit | 0.839 ** | 0.467 ** | 0.858 ** | 1 |

Note: ** $p < 0.01$.

### 3.3. Empirical Results

The ε is introduced to prevent the occurrence of weights being zero, which would otherwise lead to relevant measurements being ignored in the performance evaluation process. Table 6 illustrates the R&D efficiency scores obtained from the output-oriented relational model (1) as well as their corresponding ranks. It is clear that Table 6 provides the R&D efficiency scores for the whole system ($E_k^s$) and the period efficiencies ($E_k^{(1)}$, $E_k^{(2)}$, $E_k^{(3)}$) obtained via formulae (2) and (3), respectively.

**Table 6.** R&D efficiency measures of food manufacturing companies in China.

| DMU | Company | Sector | System | Rank | 2019 | Rank | 2020 | Rank | 2021 | Rank |
|---|---|---|---|---|---|---|---|---|---|---|
| 1 | Yunnan Energy Investment | Organic food | 0.6449 | 5 | 0.969 | 3 | 0.73 | 8 | 0.851 | 4 |
| 2 | Sanquan Food | General foods | 0.2351 | 12 | 0.437 | 12 | 0.455 | 12 | 0.397 | 13 |
| 3 | Fuling Preserved Pickles | General foods | 0.9387 | 2 | 0.865 | 4 | 1 | 1 | 0.995 | 3 |
| 4 | Anhui Jinhe Industrial | Additives | 0.0729 | 24 | 0.071 | 27 | 0.072 | 27 | 0.077 | 26 |
| 5 | Ke Ming Food | General foods | 0.1798 | 17 | 0.114 | 24 | 0.226 | 22 | 0.236 | 20 |
| 6 | YanKershop | General foods | 0.0557 | 27 | 0.082 | 26 | 0.06 | 28 | 0.065 | 27 |
| 7 | Zhuangyuan Pasture | Dairy | 0.3199 | 10 | 0.398 | 13 | 0.741 | 7 | 0.636 | 7 |
| 8 | New Hope Dairy | Dairy | 0.2127 | 16 | 0.283 | 18 | 0.256 | 19 | 0.308 | 16 |
| 9 | Garden Biochem | Food medicine | 0.159 | 18 | 0.35 | 16 | 0.396 | 15 | 0.306 | 17 |
| 10 | Angel Yeast | Fermented products | 0.0635 | 26 | 0.057 | 28 | 0.072 | 26 | 0.064 | 28 |
| 11 | Hengshun Vinegar | Condiments | 0.2128 | 15 | 0.383 | 14 | 0.488 | 11 | 0.392 | 14 |
| 12 | Terun | Dairy | 0.6072 | 6 | 0.754 | 6 | 0.831 | 5 | 0.776 | 5 |
| 13 | Sanyuan Food | Dairy | 0.4724 | 9 | 0.658 | 9 | 0.565 | 10 | 0.567 | 8 |
| 14 | Bright Dairy | Dairy | 0.9254 | 3 | 0.823 | 5 | 0.955 | 4 | 1 | 1 |
| 15 | Star Lake Bioscience | Additives | 0.1398 | 20 | 0.36 | 15 | 0.324 | 16 | 0.29 | 18 |
| 16 | MeiHua Holdings Group | Amino acids | 0.5062 | 8 | 0.695 | 7 | 0.674 | 9 | 0.503 | 9 |
| 17 | Milkland | Dairy | 0.2698 | 11 | 0.5 | 11 | 0.439 | 13 | 0.485 | 10 |
| 18 | Yili Dairy | Dairy | 0.9993 | 1 | 1 | 1 | 1 | 1 | 1 | 1 |
| 19 | Snowy Sky | Condiments | 0.1183 | 21 | 0.256 | 20 | 0.261 | 18 | 0.2 | 23 |
| 20 | Apple Flavor & Fragrance Group | Spice and additives | 0.1444 | 19 | 0.295 | 17 | 0.29 | 17 | 0.309 | 15 |
| 21 | Qianhe Condiment and Food | Condiments | 0.5267 | 7 | 0.685 | 8 | 0.742 | 6 | 0.73 | 6 |
| 22 | Guangzhou Restaurant Group | General foods | 0.1067 | 22 | 0.214 | 22 | 0.178 | 24 | 0.247 | 19 |
| 23 | Shengda Bio-Pharm | Additives | 0.0716 | 25 | 0.151 | 23 | 0.188 | 23 | 0.162 | 24 |
| 24 | Haitian Flavouring | Condiments | 0.2157 | 14 | 0.235 | 21 | 0.229 | 21 | 0.216 | 21 |
| 25 | Jingshen Salt & Chemical Industry | Condiments | 0.1029 | 23 | 0.27 | 19 | 0.25 | 20 | 0.206 | 22 |
| 26 | AnKee | Condiments | 0.2295 | 13 | 0.528 | 10 | 0.431 | 14 | 0.46 | 12 |
| 27 | Vland Biotech | Fermented products | 0.0443 | 28 | 0.088 | 25 | 0.079 | 25 | 0.106 | 25 |
| 28 | Toly Bread | Bakery | 0.7294 | 4 | 1 | 1 | 0.971 | 3 | 0.485 | 10 |
| | Average | | 0.3323 | | 0.4471 | | 0.4607 | | 0.4309 | |

Additionally, we used Spearman's rank-order correlation to check the relationship of non-parametric measures with the ranked variables in three years [54]. Based on the three-year efficiency rankings (see Table 6), we obtained the non-parametric rank correlation coefficients, in Table 7. The results show a significant positive correlation between the

rankings at the significance level of 0.01. In addition, the correlation coefficients for the rankings are all greater than 0.9, indicating a strong correlation between the efficiency rankings within three years.

**Table 7.** Spearman's rank correlation test results for rankings in three years.

|  | **System** | **2019** | **2020** | **2021** |
|---|---|---|---|---|
| System | 1 |  |  |  |
| 2019 | 0.940 ** | 1 |  |  |
| 2020 | 0.945 ** | 0.963 ** | 1 |  |
| 2021 | 0.952 ** | 0.942 ** | 0.948 ** | 1 |

Note: ** $p < 0.01$.

We find that none of the food manufacturing companies had an effective R&D efficiency ($E_k^s = 1$) for their systems during the three years. In particular, only the DMU18 (Yili Dairy) had an R&D efficiency of 0.999, which is close to being fully efficient. This indicates that Yili Dairy has maximised technological innovation, resulting in a high-level input–output efficiency without the need for input or output adjustments. This was closely followed by DMU3 (Fuling Preserved Pickles) and 14 (Bright Dairy) with high R&D efficiency. On the other hand, DMUs 6 (YanKershop), 10 (Angel Yeast) and 27 (Vland Biotech) have the lowest R&D efficiency scores.

In 2019, the company's R&D levels for DMUs 18 (Yili Dairy) and 28 (Toly Bread) were highly efficient, with efficiency scores of 1. The top two companies that obtained the highest system efficiency, Yili Dairy and Fuling Preserved Pickles, also achieved the top two positions in R&D efficiency during 2020. Besides, the most efficient R&D companies in the period 2021 are two dairy companies, Yili Dairy and Bright Dairy. Similarly, the companies with the lowest R&D efficiency were the same as those with the lowest system efficiency in each period (DMUs 6, 10, and 27).

The last row of Table 6 illustrates the average efficiency scores of the whole system and each of the three years. The mean of the system R&D efficiency score is 0.3323, which indicates that the industry's R&D performance is at a relatively low level between 2019 and 2021. Precisely, the period from 2019 to 2020 showed slight growth in the average R&D efficiency score, reaching the peak at 0.4607 in 2020. However, the average score fell again to 0.4309 in 2021.

It is worth mentioning that dairy companies were at a high level, while those of the spice and fermented product companies were relatively low. In particular, three of the six companies with moderately high system efficiency scores (efficiency score greater than or equal to 0.6) are the dairy manufacturing companies; these are the DMUs 12, 14 and 18. Additionally, four of the five companies with low system efficiency scores (score less than or equal to 0.1) are the additives and fermented product manufacturers; these are the DMUs 4, 10, 23 and 27. There is a wide gap in R&D efficiency among listed companies in the food manufacturing industry, which implies that the technological innovation capacity needs to be improved. Testing efficiency differences both by year and by sector per year on the selected time periods, it seems that there are no efficiency differences between them (Table 8). However, the little *p*-value in year 2021 for sectorial differences may indicate that there is a need for a greater sample in order to potentially highlight statistically significant differences between sectors.

**Table 8.** Kruskal–Wallis for the acquired efficiency scores: (a) between years, (b) between years by sector.

| | $\chi^2$ | df | P | $\varepsilon^2$ |
|---|---|---|---|---|
| a Between years | | | | |
| Eff. scores | 0.0696 | 2 | 0.966 | $8.38 \times 10^{-4}$ |
| b Between years by sector | | | | |
| 2019 | 7.29 | 5 | 0.210 | 0.270 |
| 2020 | 7.58 | 5 | 0.810 | 0.281 |
| 2021 | 11.06 | 5 | 0.070 | 0.409 |

## 4. Discussion

The food manufacturing sector is at the bottom of the industry in terms of efficiency of investment in R&D, which is consistent with previous studies [25,26]. To the best of our knowledge, this is the first-time dynamic DEA has been applied in the food sector, while this methodology is mostly used in the technology sector, as mentioned in the literature review section. A similar approach was used from Halaskova et al. (2021) to measure the efficiency of public and private sectors. In their paper, they explored two different time-periods between 2010/2013 and 2014/2017, considering each of the three-year periods as unified systems [55]. However, as Kao (2014) stated ignoring the operations of the component divisions (i.e., periods) may obtain misleading results [56].

Additionally, there is a severe differentiation in the R&D capabilities of food manufacturing companies, mainly in the form of outstanding R&D capabilities of some of the top companies, while most of the remaining companies are at the stage of weaker R&D capabilities. For example, China's current innovative tea products have a low barrier to entry and lack technical barriers, resulting in a highly competitive and homogenized market. After Heytea pioneered the cheese tea product, other brands scrambled to imitate and copy the recipe. However, as there have been no significant products' differentiation, it has been challenging for brands to gain an advantage in the industry. The spillover effects mentioned in the literature review could explain this phenomenon [27]. Technology development requires significant investment and carries significant risks as well, and when the "technology spillovers" occurs, other companies are significantly less motivated to innovate. This is because competitors can acquire the technology of the leading company at no cost and invest in the production of new products [27]. Although new technologies can benefit the food market, relying on "technology spillovers" is not a sustainable solution. If the overall R&D capacity cannot be enhanced, then the economic growth of the whole industry will be affected. Research and development in science and technology is a booster and strong support for the development of the food industry while seeking to meet new consumer demands such as convenience, nutrition, and health.

When the melamine-tainted milk powder incident came to light in 2008, the whole domestic dairy industry suffered a severe turbulence [57]. At that time, consumers were willing to increase their expenditures on foreign brands rather than trust domestic products anymore. The breach of trust by some dairy companies led to a severe winter for the entire domestic dairy industry. Following this, the Chinese government has reformed the management of dairy products and related laws to raise technical standards and improve the safety of dairy products. For example, the regulation of the entire dairy chain has been strengthened, and the technical requirements and specific standards relating to dairy products have been made more operational [58]. In the wake of the scandal, the dismal performance of dairy companies also serves as a wake-up call to companies that food safety is always uppermost in the minds of consumers. Strict external regulation coupled with internal corporate awakening has led to a decade of accelerated investment in research and development by dairy companies in an effort to regain consumer trust and confidence.

One of the most outstanding R&D performers is Yili Dairy. Yili Dairy has created the first three-tier R&D system in the industry, with R&D centres at all levels empowering and supporting each other. Moreover, innovation centres have been built in seven locations around the world, attracting a large number of dairy research talents and providing intellectual support for the company's research and technological innovation. In 2020, Yili topped the list of "Top 10 Food Innovation Companies in China". As of December 2021, Yili ranked second among the top 10 companies in the world's dairy industry in terms of the total number of global patent applications and the total number of invention applications [59]. Against this backdrop, it makes sense that the DEA measurements show that dairy companies excel in R&D efficiency and Yili has a strong performance in R&D in the food manufacturing industry. Moreover, this reflects the fact that consumers will always put food quality and safety at the forefront of their minds.

Additives and fermented products are typical applications in the field of biotechnology [60]. The results show that R&D in food biotechnology companies is less efficient than in the technology sector, which is in line with Fu's view [61]. Based on the dataset, we found that additives and fermented manufacturers invest a lot of research staff and R&D expenditure, but do not get the expected returns. Possible explanations for this are: (a) the poor capability of R&D labour, and (b) the low utilisation of R&D expenditure, which makes it more challenging to translate R&D activities into economic benefits. Another possibility is the mismatch between R&D investment and the company's own needs, resulting in a waste of human and financial resources and thus inefficient R&D. Additionally, a study by Wang et al. [62] suggested that R&D results in biotechnology have a lagging effect. We speculate that there is a high probability that food companies producing biotechnology will be in a similar situation. It may require a period of time after R&D investment before technological innovation can be injected into their operations as a growth driver.

A limitation of this study is that the selection of input and output factors is based on literature reviews and does not use objective methods such as regression. The reason behind this is the incomplete disclosure of non-financial data relating to R&D performance inputs and outputs, such as the number of patents, in the annual statements of the listed companies, which limits the selection of indicators. Publicly available annual reports do not disclose R&D information until after 2018, causing limited access to R&D information. Following the discussion in Djordjevic et al. (2021), we have implemented the Pearson correlation to firstly ensure that the input/output classification is robust and justified and, secondly, to avoid considering any indicators which are not significantly correlated at 99% confidence interval to other indicators [63].

Moreover, an extended time period could have been analyzed, considering technological change through the years following Window DEA principles [64], given the existence of an appropriate dataset that reflects this period

Another constraint of this paper was the external influence of COVID-19 pandemic in the R&D sector. Although an effort for clarifying efficiency differences among years was made through a paired-Wilcoxon test, no significant differences were obtained from this process. Similar results for the R&D performance were obtained from Yi et al. [65], where they concluded that more funding is needed for implications of innovative applications in the Chinese food sector. Despite the fact that this survey was concerning R&D performance in COVID-19 period, there was no clue for the internal structure of the analysed firms, and researchers had applied the Tobit model to further assess inefficiency causes. Although Jin et al. [66] reported that COVID-19 had a negative impact on high tech companies, the influence of the pandemic on technological skills of employees was positive. From another perspective, R&D investments made before the pandemic, were deemed as a resistance factor for minimising companies' losses in the COVID-19 era, highlighting the importance of such investments in extreme case scenarios such as the pandemic period [67].

Due to the fact that COVID-19 had a negative impact on enterprises' profitability, R&D expenses in the food sector have been minimised or annihilated [68]. Assessing R&D performance in the food sector during the COVID-19 era demonstrates that in a

short-term analysis there are no apparent differences. Thus, this paper aims to provide preliminary results of COVID-19's impact on R&D performance of the agricultural/food sector; however, it is essential to acquire future data to gain more meaningful insights on this phenomenon.

## 5. Conclusions

### 5.1. Contributions & Practical Applications

Drawing on Kao's dynamic relational model [20], this study analyses the R&D efficiency of 28 listed companies in China's food manufacturing industry from 2019 to 2021, to assess the level of R&D performance of the industry. In this paper, a three-year dynamic R&D system is also developed, including inputs, outputs, and carry-over factors, in order to calculate R&D efficiency scores for the whole system and each period, respectively.

Against the backdrop of low overall R&D efficiency, there is a need to leverage the technological strengths of large enterprises in the food manufacturing industry while mobilising the R&D enthusiasm of SMEs [69]. To narrow the R&D efficiency gap between enterprises, the core competitiveness of the entire food manufacturing industry can be improved through synergistic development.

At the corporate level, food manufacturing companies should place emphasis on R&D activities and invest in R&D resources to gain a competitive advantage over their competitors. The food manufacturing industry needs to strengthen its technological research and development activities and promote technological transformation to develop products that meet the needs of consumers, investors, and regulators. In addition, while approaching advanced technologies, companies need to avoid technological dependence and achieve breakthroughs in independent innovation. For example, in order to develop a probiotic with its intellectual property rights, Yili Dairy went through rigorous experimental screening to create a probiotic strain that is beneficial to the intestinal health of Chinese people, Bifidobacterium lactis BL-99. In terms of the use of R&D resources, on the one hand, the inputs should be controlled to ensure that they are in line with the development status of the company. On the other hand, project monitoring and other tools (e.g., regular audits) can be used to measure whether the inputs are used effectively.

In addition, companies need to strengthen the management of R&D talent and improve talent incentive policies. To attract high-level innovative talent, enterprises need to create a good employment environment and provide preferential policies, such as additional opportunities for R&D staff, career development planning and regular training. In addition, as innovation seekers, food companies should cooperate with research institutes or higher education institutions. The enterprises can make full use of the research resources provided in order to solve their own bottleneck problem of insufficient scientific and technological resources. Meanwhile, universities and research institutes may also need to use the plants and processing equipment of enterprises as a pilot test or as practice bases. This type of mutually beneficial cooperation can positively affect R&D resources management, by saving company funds and creating the conditions for developing new products.

The government should also improve its policy and financial support so as to optimise the R&D environment. Tax relief policy can reduce the R&D costs of enterprises with greater intensity and direct the flow of production factors to the innovation sector. Therefore, the government should continue to implement tax relief policies so that enterprises will have more confidence to invest in research and innovation to achieve quality development.

Meanwhile, small, and medium-sized enterprises (SMEs) in the food industry often lack capital and talent and can be subjected to technological suppression by large companies. Therefore, there is little incentive for R&D activities. The government needs to allocate more resources to SMEs; for example, by widening their access to finance to alleviate the financial constraints on research and development. Companies that have an extensive unproductive time period in transforming their R&D results also need to be offered extra support, such as financial subsidies or establishing awards for research results as an additional incentive.

Extra support can prevent such companies from being sapped of their own enthusiasm for research and development due to time and risk issues.

The government needs to improve the food patent protection system and create a business environment that respects innovation. As mentioned above, R&D activity in the food industry tends to be concentrated in some companies due to the "spillover" effect, with the remainder surviving by copying and imitating. Patent law should be able to effectively protect research and development achievements as well as the legitimate rights and interests of enterprises. At the same time, this will encourage research and development activities, motivate other enterprises to carry out inventions and further promote the progress and innovation of the food industry.

### *5.2. Future Research Directions*

In addition, this study considers the whole system as a "black box" without fully acknowledging the internal mechanisms within the R&D activities [20,70]. To this end, it does not assess the various divisions in R&D activities and the reasons that lead to their inefficiency. Uncovering the dynamic network structure of R&D activities is an intriguing future research direction that could be explored. The dynamic network DEA approach of Omrani and Soltanzadeh [36] could be implemented for R&D efficiency scores estimation over time, considering the relationship between system and divisions [34].

**Author Contributions:** Conceptualization, S.M.; methodology, S.M., M.D.K. and L.S.K.; software, S.M. and M.D.K.; validation, M.D.K. and G.V.; formal analysis, S.M.; investigation, L.S.K.; resources, L.S.K.; data curation, S.M.; writing—original draft preparation, S.M. and M.D.K.; writing—review and editing, M.D.K., G.V. and L.S.K.; visualization, L.S.K.; supervision, M.D.K., L.S.K. and G.V. All authors have read and agreed to the published version of the manuscript.

**Funding:** This research received no external funding.

**Institutional Review Board Statement:** Not applicable.

**Informed Consent Statement:** Not applicable.

**Data Availability Statement:** The data presented in this study are embodied in the Appendix A.

**Conflicts of Interest:** The authors declare no conflict of interest.

## Appendix A

**Table A1.** Relevant dataset.

| Company | Stock Code | Year | Inputs | | Outputs | | Carry-Over |
| | | | R&D Staff | R&D Expenditure | Operating Income | Net Profit | Capital Stock |
|---|---|---|---|---|---|---|---|
| 1 | 002053 | 2018 | | | | | 558,329,336 |
| | 002053 | 2019 | 48 | 9,068,433 | 1,933,137,924 | 286,313,835 | 760,978,566 |
| | 002053 | 2020 | 220 | 11,284,582 | 1,990,278,508 | 259,553,422 | 760,978,566 |
| | 002053 | 2021 | 217 | 5,823,097 | 2,258,831,335 | 270,439,271 | 760,978,566 |
| 2 | 002216 | 2018 | | | | | 809,664,717 |
| | 002216 | 2019 | 109 | 149,926,685 | 5,985,722,254 | 219,150,712 | 799,258,226 |
| | 002216 | 2020 | 106 | 118,442,081 | 6,926,082,823 | 767,687,663 | 799,258,226 |
| | 002216 | 2021 | 151 | 96,269,725 | 6,943,439,865 | 639,942,313 | 879,184,048 |
| 3 | 002507 | 2018 | | | | | 789,357,241 |
| | 002507 | 2019 | 21 | 25,962,179 | 1,989,593,123 | 605,141,874 | 789,357,241 |
| | 002507 | 2020 | 21 | 21,494,619 | 2,272,746,599 | 777,105,783 | 789,357,241 |
| | 002507 | 2021 | 22 | 25,048,460 | 2,518,647,389 | 741,958,457 | 887,630,022 |
| 4 | 002597 | 2018 | | | | | 558,768,374 |
| | 002597 | 2019 | 434 | 136,639,529 | 3,971,856,106 | 808,356,014 | 558,771,351 |
| | 002597 | 2020 | 488 | 120,403,962 | 3,666,246,520 | 718,521,570 | 560,903,311 |
| | 002597 | 2021 | 516 | 182,871,407 | 5,845,322,601 | 1,176,448,711 | 560,913,735 |
| 5 | 002661 | 2018 | | | | | 331,930,298 |
| | 002661 | 2019 | 99 | 41,854,132 | 3,033,973,309 | 206,668,169 | 328,808,450 |
| | 002661 | 2020 | 94 | 27,400,192 | 3,957,752,136 | 292,756,153 | 334,760,450 |
| | 002661 | 2021 | 95 | 28,467,594 | 4,326,648,257 | 67,367,507 | 337,010,083 |

**Table A1.** *Cont.*

| Company | Stock Code | Year | Inputs | | Outputs | | Carry-Over |
|---|---|---|---|---|---|---|---|
| | | | R&D Staff | R&D Expenditure | Operating Income | Net Profit | Capital Stock |
| 6 | 002847 | 2018 | | | | | 124,000,000 |
| | 002847 | 2019 | 68 | 26,900,715 | 1,399,275,041 | 127,613,274 | 128,400,000 |
| | 002847 | 2020 | 173 | 51,500,758 | 1,958,851,487 | 242,120,751 | 129,360,000 |
| | 002847 | 2021 | 169 | 55,190,165 | 2,281,504,302 | 154,363,226 | 129,360,000 |
| 7 | 002910 | 2018 | | | | | 187,340,000 |
| | 002910 | 2019 | 35 | 9,461,944 | 813,554,461 | 51,321,172 | 190,680,600 |
| | 002910 | 2020 | 15 | 9,103,148 | 739,820,698 | 10,453,468 | 233,680,600 |
| | 002910 | 2021 | 16 | 9,246,922 | 1,021,431,542 | 53,533,056 | 232,381,032 |
| 8 | 002946 | 2018 | | | | | 768,339,599 |
| | 002946 | 2019 | 123 | 69,754,667 | 5,674,953,670 | 251,445,167 | 853,710,666 |
| | 002946 | 2020 | 150 | 74,966,721 | 6,748,631,857 | 289,434,618 | 853,710,666 |
| | 002946 | 2021 | 146 | 90,797,649 | 8,966,872,398 | 341,261,734 | 867,271,477 |
| 9 | 300401 | 2018 | | | | | 479,288,315 |
| | 300401 | 2019 | 97 | 42,757,359 | 718,384,536 | 343,706,533 | 479,288,315 |
| | 300401 | 2020 | 99 | 29,736,199 | 614,894,441 | 272,264,656 | 551,007,557 |
| | 300401 | 2021 | 137 | 68,545,827 | 1,117,099,893 | 510,007,726 | 551,007,557 |
| 10 | 600298 | 2018 | | | | | 824,080,943 |
| | 600298 | 2019 | 591 | 333,462,090 | 7,652,754,552 | 939,880,334 | 824,080,943 |
| | 600298 | 2020 | 640 | 386,191,432 | 8,933,035,778 | 1,422,132,783 | 824,080,943 |
| | 600298 | 2021 | 730 | 475,198,936 | 10,675,333,008 | 1,321,484,377 | 832,860,943 |
| 11 | 600305 | 2018 | | | | | 783,559,400 |
| | 600305 | 2019 | 124 | 52,943,612 | 1,832,193,611 | 330,267,318 | 783,559,400 |
| | 600305 | 2020 | 124 | 57,769,207 | 2,014,309,859 | 320,078,453 | 1,002,956,032 |
| | 600305 | 2021 | 151 | 78,542,779 | 1,893,347,830 | 118,332,057 | 1,002,956,032 |
| 12 | 600419 | 2018 | | | | | 207,114,418 |
| | 600419 | 2019 | 10 | 3,182,198 | 1,626,592,714 | 142,028,807 | 207,114,418 |
| | 600419 | 2020 | 13 | 5,223,872 | 1,767,673,596 | 153,602,820 | 268,599,337 |
| | 600419 | 2021 | 18 | 6,498,985 | 2,109,258,101 | 160,959,099 | 320,190,246 |
| 13 | 600429 | 2018 | | | | | 1,497,557,426 |
| | 600429 | 2019 | 60 | 20,355,603 | 8,150,710,057 | 159,042,286 | 1,497,557,426 |
| | 600429 | 2020 | 71 | 35,327,399 | 7,353,344,572 | 15,400,349 | 1,497,557,426 |
| | 600429 | 2021 | 73 | 117,654,695 | 7,730,723,573 | 208,678,748 | 1,497,557,426 |
| 14 | 600597 | 2018 | | | | | 1,224,487,509 |
| | 600597 | 2019 | 100 | 68,140,427 | 22,563,236,819 | 682,452,363 | 1,224,487,509 |
| | 600597 | 2020 | 94 | 72,844,897 | 25,222,715,966 | 785,141,962 | 1,224,487,509 |
| | 600597 | 2021 | 101 | 89,259,433 | 29,205,992,515 | 566,893,573 | 1,378,640,863 |
| 15 | 600866 | 2018 | | | | | 645,393,465 |
| | 600866 | 2019 | 284 | 53,550,926 | 1,049,609,531 | 149,552,368 | 739,019,166 |
| | 600866 | 2020 | 312 | 58,679,169 | 1,116,277,268 | 148,710,264 | 739,019,166 |
| | 600866 | 2021 | 356 | 68,877,956 | 1,235,046,858 | 106,469,881 | 739,019,166 |
| 16 | 600873 | 2018 | | | | | 3,108,175,038 |
| | 600873 | 2019 | 98 | 434,643,905 | 14,553,547,455 | 1,003,557,478 | 3,104,289,638 |
| | 600873 | 2020 | 115 | 468,201,631 | 17,049,514,475 | 1,005,432,475 | 3,100,021,848 |
| | 600873 | 2021 | 219 | 649,213,308 | 22,836,890,325 | 2,376,147,704 | 3,098,619,928 |
| 17 | 600882 | 2018 | | | | | 409,762,045 |
| | 600882 | 2019 | 46 | 22,304,115 | 1,744,349,052 | 19,229,864 | 409,357,045 |
| | 600882 | 2020 | 60 | 38,860,661 | 2,846,807,171 | 73,984,474 | 409,309,045 |
| | 600882 | 2021 | 76 | 40,090,274 | 4,478,305,562 | 193,769,127 | 516,210,147 |
| 18 | 600887 | 2018 | | | | | 6,078,127,608 |
| | 600887 | 2019 | 411 | 541,803,036 | 90,009,132,852 | 6,950,726,155 | 6,096,378,858 |
| | 600887 | 2020 | 453 | 487,099,849 | 96,523,963,250 | 7,098,938,695 | 6,082,624,833 |
| | 600887 | 2021 | 461 | 601,017,082 | 110,143,986,386 | 8,732,025,624 | 6,400,130,918 |
| 19 | 600929 | 2018 | | | | | 917,751,148 |
| | 600929 | 2019 | 453 | 87,867,120 | 2,272,012,599 | 163,294,814 | 917,751,148 |
| | 600929 | 2020 | 480 | 85,470,249 | 2,164,477,645 | 156,301,166 | 917,751,148 |
| | 600929 | 2021 | 772 | 193,944,936 | 4,780,264,152 | 441,029,995 | 1,350,168,875 |
| 20 | 603020 | 2018 | | | | | 320,000,000 |
| | 603020 | 2019 | 124 | 33,018,395 | 2,474,657,721 | 170,251,807 | 320,000,000 |
| | 603020 | 2020 | 130 | 34,034,181 | 2,668,255,090 | 192,710,292 | 320,000,000 |
| | 603020 | 2021 | 140 | 39,910,113 | 3,344,556,763 | 232,993,957 | 383,237,774 |
| 21 | 603027 | 2018 | | | | | 326,202,714 |
| | 603027 | 2019 | 37 | 39,482,609 | 1,355,147,204 | 198,253,971 | 465,850,722 |
| | 603027 | 2020 | 45 | 45,221,539 | 1,693,273,982 | 205,801,040 | 665,675,318 |
| | 603027 | 2021 | 45 | 55,389,753 | 1,925,286,294 | 221,401,595 | 798,782,158 |

**Table A1.** *Cont.*

| Company | Stock Code | Year | Inputs | | Outputs | | Carry-Over |
| | | | R&D Staff | R&D Expenditure | Operating Income | Net Profit | Capital Stock |
|---|---|---|---|---|---|---|---|
| 22 | 603043 | 2018 | | | | | 403,996,184 |
| | 603043 | 2019 | 292 | 61,337,012 | 3,028,699,726 | 383,438,304 | 403,996,184 |
| | 603043 | 2020 | 397 | 77,382,774 | 3,287,486,223 | 464,237,089 | 403,996,184 |
| | 603043 | 2021 | 346 | 77,446,242 | 3,889,924,382 | 564,886,983 | 565,594,658 |
| 23 | 603079 | 2018 | | | | | 112,000,000 |
| | 603079 | 2019 | 137 | 27,901,281 | 519,149,812 | 37,759,207 | 112,000,000 |
| | 603079 | 2020 | 135 | 35,685,966 | 867,314,753 | 189,972,212 | 171,188,958 |
| | 603079 | 2021 | 145 | 39,643,033 | 789,732,755 | 71,822,072 | 171,188,958 |
| 24 | 603288 | 2018 | | | | | 2,700,369,340 |
| | 603288 | 2019 | 426 | 587,425,291 | 19,796,889,800 | 5,356,242,595 | 2,700,369,340 |
| | 603288 | 2020 | 513 | 711,748,663 | 22,791,873,936 | 6,409,030,014 | 3,240,443,208 |
| | 603288 | 2021 | 599 | 771,919,702 | 25,004,031,043 | 6,671,470,526 | 4,212,576,170 |
| 25 | 603299 | 2018 | | | | | 559,440,000 |
| | 603299 | 2019 | 368 | 89,000,000 | 4,190,385,022 | 260,205,816 | 775,730,854 |
| | 603299 | 2020 | 368 | 89,230,000 | 3,937,297,486 | 152,027,531 | 774,379,748 |
| | 603299 | 2021 | 392 | 115,736,800 | 4,761,367,450 | 340,989,147 | 772,926,545 |
| 26 | 603696 | 2018 | | | | | 168,000,000 |
| | 603696 | 2019 | 33 | 13,180,385 | 421,296,739 | 42,797,000 | 235,200,000 |
| | 603696 | 2020 | 37 | 10,572,417 | 420,400,962 | 52,817,550 | 235,200,000 |
| | 603696 | 2021 | 33 | 13,775,589 | 548,965,239 | 45,392,683 | 235,200,000 |
| 27 | 603739 | 2018 | | | | | 116,000,000 |
| | 603739 | 2019 | 196 | 74,997,960 | 846,777,963 | 89,391,435 | 154,667,000 |
| | 603739 | 2020 | 216 | 79,913,467 | 960,249,354 | 122,304,777 | 154,667,000 |
| | 603739 | 2021 | 264 | 100,936,653 | 1,150,823,565 | 150,097,268 | 252,084,840 |
| 28 | 603866 | 2018 | | | | | 470,626,000 |
| | 603866 | 2019 | 24 | 8,842,772 | 5,643,709,760 | 683,358,392 | 658,876,400 |
| | 603866 | 2020 | 34 | 11,295,700 | 5,963,004,181 | 882,839,002 | 680,152,702 |
| | 603866 | 2021 | 89 | 20,645,747 | 6,335,381,672 | 763,265,674 | 952,213,783 |

**Table A2.** Distribution characteristics of selected indicators per year.

| Period | Role | Indicator | Year | Average | St.Dev. | Max. | Min. |
|---|---|---|---|---|---|---|---|
| 2019 | Carry-over | Capital stock | 2018 | 909,987,897 | 1,233,167,228 | 6,078,127,608 | 112,000,000 |
| | Input | R&D expenditure | 2019 | 109,491,585 | 160,342,599 | 587,425,291 | 3,182,198 |
| | Input | R&D labour | 2019 | 173 | 164 | 591 | 10 |
| | Carry-over | Capital stock | 2019 | 947,126,410 | 1,224,655,656 | 6,096,378,858 | 112,000,000 |
| | Output | Operating revenue | 2019 | 7,687,546,550 | 17,074,535,941 | 90,009,132,852 | 421,296,739 |
| | Output | Net profit | 2019 | 737,908,824 | 1,569,025,767 | 6,950,726,155 | 19,229,864 |
| 2020 | Carry-over | Capital stock | 2019 | 947,126,410 | 1,224,655,656 | 6,096,378,858 | 112,000,000 |
| | Input | R&D expenditure | 2020 | 116,253,048 | 174,284,845 | 711,748,663 | 5,223,872 |
| | Input | R&D labour | 2020 | 200 | 180 | 640 | 13 |
| | Carry-over | Capital stock | 2020 | 990,183,108 | 1,246,656,935 | 6,082,624,833 | 129,360,000 |
| | Output | Operating revenue | 2020 | 8,444,877,879 | 18,390,488,392 | 96,523,963,250 | 420,400,962 |
| | Output | Net profit | 2020 | 838,620,037 | 1,708,771,917 | 7,098,938,695 | 10,453,468 |
| 2021 | Carry-over | Capital stock | 2020 | 990,183,108 | 1,246,656,935 | 6,082,624,833 | 129,360,000 |
| | Input | R&D expenditure | 2021 | 147,070,102 | 207,936,664 | 771,919,702 | 5,823,097 |
| | Input | R&D labour | 2021 | 231 | 213 | 772 | 16 |
| | Carry-over | Capital stock | 2021 | 1,095,924,855 | 1,356,906,255 | 6,400,130,918 | 129,360,000 |
| | Output | Operating revenue | 2021 | 9,932,821,966 | 20,985,555,792 | 110,143,986,386 | 548,965,239 |
| | Output | Net profit | 2021 | 972,979,718 | 1,983,880,346 | 8,732,025,624 | 45,392,683 |

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
