# Peer review of "R&D Performance Evaluation in the Chinese Food Manufacturing Industry Based on Dynamic DEA in the COVID-19 Era"

_agriculture, doi:10.3390/agriculture12111938_

Round 1

Reviewer 1 Report

Recommendations:

- For the paragraph (590-598) don't use self-citation, insert another reference.

- In line 593, insert "the listed";

- In line 596, correct "analysed";

- In line 598, correct "refrlects";

Author Response

We thank the reviewer for expressing a general appreciation of the topic this paper addresses. 

The comments provided have helped us in producing an adapted and improved version of the document.  

We think we have been able to meet your expectations regarding us addressing your stated concerns, by making adjustments to the paper.

The changes to the manuscript itself can be found via the track changes system of MS Word. 

We would like to thank you again for your valuable time in reviewing our manuscript!

Sincerely yours,
The authors, 14/11/2022.

Reviewer 2 Report

The objective of the current study was to measure and evaluate the level of research and development (R&D) performance in the Chinese food manufacturing industry by implementing the Dynamic Data Envelopment Analysis (DDEA) technique. The manuscript is well written and organized. However, it is very long and looks like a review article. some major concerns should be clarified.

1.      The sections of introduction and literature review should be merged in one section and summarized to 2-3 pages.

2.      The authors should clarify why the Chinese food manufacturing companies were chosen in the current study?

3.      A sub-section of results analysis should be included in the  methodology.

4.      The methodology of companies selection should be clarified to prevent any bias or error.  

5.      The results are not well presented and not connected to each other. and no appropriate discussion and the results should be compared with other studies. More studies should be added

6.      Statistical analysis should be added to tables 3 to identify the significant differences between the companies as well as the sectors.

7.      Sections of conclusion should be summarized to 1-2 paragraphs 

8.      Large numbers of references were used.

Reviewer 3 Report

Dear editor and authors,

Thank you for the opportunity to read this interesting and current article aimed to analyze the R&D performance evaluation in the Chinese food manufacturing industry based on dynamic DEA in the COVID-19 era.

The paper is complete in all its scientific elements. The introduction and the literature review are relevant to the topic of the research. Similarly, the methodology adopted could be easily emulated for further research in this field. The topic is very important is a theme that deserves more attention.

In the same way the results are clear, but the importance and practical application of the work needs to be highlighted, preferably under a separate heading . To conclude, the manuscript is sustainable for agriculture journal and the paper enriches the scientific literature since the results are relevant.

Author Response

We thank the reviewer for their fruitful comments provided through the reviewing process. 

The comments provided have helped us in producing an adapted and improved version of the document so that in particular we have been able to better explain the contributions of this manuscript.

We think we have been able to meet your expectations regarding us addressing your stated concerns, by making adjustments to the manuscript as you kindly indicated.

The changes to the manuscript itself can be found via the track changes system of MS Word. 

We would like to thank you again for your valuable time in reviewing our manuscript!

Sincerely yours,
The authors, 14/11/2022.

Round 2

Reviewer 2 Report

The manuscript is significantly improved and can be accepted in this form.